# Fine Particulate Matter Exposure Levels in Patients with Normal-Tension Glaucoma and Primary Open-Angle Glaucoma: A Population-Based Study from Taiwan

**DOI:** 10.3390/ijerph19074224

**Published:** 2022-04-01

**Authors:** Ci-Wen Luo, Yun-Wei Chiang, Han-Yin Sun, Kun-Lin Yeh, Min-Wei Lee, Wen-Jun Wu, Yu-Hsiang Kuan

**Affiliations:** 1Institute of Medicine, Chung Shan Medical University, Taichung 40201, Taiwan; s0803019@gm.csmu.edu.tw (C.-W.L.); 007wu@csmu.edu.tw (W.-J.W.); 2Department of Medical Research, Chung Shan Medical University Hospital, Taichung 40201, Taiwan; 3Department of Life Sciences, National Chung-Hsing University, Taichung 40227, Taiwan; 108589@ctust.edu.tw; 4Department of Optometry, Chung Shan Medical University, Taichung 40201, Taiwan; hanying@csmu.edu.tw; 5Department of Ophthalmology, Chung Shan Medical University Hospital, Taichung 40201, Taiwan; 6Department of Veterinary Medicine, National Chung Hsing University, Taichung 40227, Taiwan; d108038002@mail.nchu.edu.tw; 7A Graduate Institute of Microbiology and Public Health, National Chung Hsing University, Taichung 40201, Taiwan; d108046001@mail.nchu.edu.tw; 8Department of Pharmacology, School of Medicine, Chung Shan Medical University, Taichung 40201, Taiwan; 9Department of Pharmacy, Chung Shan Medical University Hospital, Taichung 40201, Taiwan

**Keywords:** PM_2.5_, normal-tension glaucoma (NTG), primary open-angle glaucoma (POAG), population-based study

## Abstract

Patients with NTG or POAG with more than one outpatient or discharge diagnosis from the ophthalmology department were included in the study. These data were merged with the PM_2.5_ data from the Air Quality Monitoring Network for analysis. This was a case–control study, with 1006 participants in the NTG group and 2533 in the POAG group. To investigate fine particulate matter (PM_2.5_) exposure levels in patients with normal-tension glaucoma (NTG) and primary open-angle glaucoma (POAG), patient data were obtained from Taiwan’s Longitudinal Health Insurance Database 2000 for the 2008 to 2013 period. We used a multivariate logic regression model to assess the risk for each participant. The PM_2.5_ exposure levels were divided into four groups: <25th percentile (Q1), <617 μg/mm^3^; 25th to 50th percentile (Q2), 617 to 1297 μg/mm^3^; 50th to 75th percentile (Q3), 1297 to 2113 μg/mm^3^; and >75th percentile (Q4), >2113 μg/mm^3^. The results are expressed in terms of odds ratio (OR) and 95% CI. A multiple logistic regression was used to compare the results of the NTG group with those of the POAG group. Compared with the PM_2.5_ Q1 level, the OR of the PM_2.5_ Q2 level was 1.009 (95% CI 0.812–1.254), the PM_2.5_ Q3 level was 1.241 (95% CI 1.241–1.537, *p* < 0.05), and the PM_2.5_ Q4 level was 1.246 (95% CI 1.008–1.539, *p* < 0.05). Our research reveals that compared with POAG, the risk of developing NTG is more closely related with PM_2.5_ exposure, and PM_2.5_ has a concentration–dose effect. It is hoped that in the future, in the clinical judgment of NTG and POAG, the level of PM_2.5_ in the environment can be taken as a risk factor.

## 1. Introduction

Glaucoma is a disease that involves the progressive loss of retinal ganglion cells, which can cause irreversible blindness in people over 60 years of age [1]. Increased intraocular pressure (IOP) leading to optic nerve damage and visual field loss is the main risk factor for glaucoma [2]. Glaucoma is the leading cause of irreversible blindness worldwide [3]. Glaucoma affects more than 70 million people worldwide, and about 10% of them are bilaterally blind [4]. Demographic surveys show that only 10% to 50% of glaucoma patients know they have glaucoma [5,6]. The subject of the current review is still NTG and primary open-angle glaucoma (POAG), which represents the continuum of open-angle glaucoma, in which a certain level of intraocular pressure (IOP) is the main risk factor for disease in POAG, and the independent factor of IOP is becoming more and more important in NTG [7]. Currently, this special form of glaucoma is called normal-tension glaucoma (NTG), also known as normal intraocular pressure glaucoma. NTG is very common in women. Hypertension and low diastolic perfusion pressure are important risk factors [8]. It was found that the patient′s progression of VF loss was associated with higher 24-hour peak IOP and greater 24-hour IOP fluctuations [9]. NTG is more common in Asia than in Western countries [10]. In clinical studies, systemic hypertension, vasospasm, acute hypotension, and neurodegenerative and psychological diseases are considered as potential risk factors for glaucoma [11,12]. It has been proposed that there is an association between NTG and migraine, and the two diseases have a potential common vascular cause [13]. The instillation of PM_2.5_ in a mouse trachea model activated the expression of Ang II, ERK1/2, and TGF-β1, causing myocardial fibrosis [14], and also inducing apoptosis and autophagy through the PI3K/AKT/mTOR signaling pathway [15]. The expression of TGF-β and the activation of the PI3K/AKT/mTOR pathway both play a role in the pathogenesis of glaucoma [15,16], resulting in a progressive loss of retinal ganglion cells (RGC) [17].

Epidemiological studies have shown that PaM_2.5_ pollution is associated with central nervous system diseases (including Alzheimer′s disease and Parkinson′s disease) that affect cranial nerves, respiratory diseases (such as asthma) and cardiovascular diseases (such as stroke and ischemia related to an increased risk of sexual heart disease) [17,18,19]. According to 23 estimates of all-cause mortality, a 10 µg/m^3^ increase in PM_2.5_ is associated with a 1.04% increase in the risk of death (95% CI 0.52% to 1.56%) [20,21]. According to the WHO standard level stratification analysis of PM_2.5_ exposure level, PM_2.5_ is an independent factor related to POAG [19]. Compared to patients without POAG, patients with POAG also have higher PM_2.5_ exposures. The absence of an association between PM_2.5_ and intraocular pressure suggests that this relationship may depend on a pressure-independent mechanism, possibly due to neurotoxicity and/or vascular effects [22]. NTG and POAG represent a continuum of open-angle glaucoma, the main difference being that a certain level of intraocular pressure is a major causative risk factor for POAG, while other factors that are not related to intraocular pressure are of increasing importance in NTG [23]. For some reasons, even if the intraocular pressure is normal, the optic nerve is easily damaged. Therefore, on the one hand, we are curious about the impact of PM_2.5_ on NTG, and on the other hand, we are also curious about the differences between the impacts of PM_2.5_ on NTG and POAG. In this study, under the assumption that PM_2.5_ affects POAG, we investigated whether exposure to air pollution, particularly PM_2.5_, would be more highly associated with NTG than POAG.

## 2. Methods

### 2.1. Data Source

The National Health Insurance Research Database (NHIRD) of Taiwan is an insurance database covering about 98% of Taiwan′s population. The source of these data is from Taiwan′s National Health Insurance (NHI) program (established by the National Health Research Institutes in 1995); the NHI program covers comprehensive medical services and retains outpatient and hospitalization records for approximately 99% of the Taiwan population. Data used in this study came from the NHIRD data subset, the 2000 longitudinal health insurance database, randomly selected from the NHIRD between 2002 and 2013, with all claims data for a total of 1 million representative beneficiaries. The NHIRD is a large and reliable source of biomedical research data not only in Taiwan, but also around the world.

### 2.2. Study Population

This study is a case–control study. The participants were patients who had received a diagnosis of NTG (ICD-9-CM: 365.12) before 2013 and did not have POAG (ICD-9-CM: 365.11), and patients who had received a diagnosis of POAG before 2013 and did not have NTG. Patients who had received a diagnosis of either NTG or POAG before 2008 were excluded. Both the patients with NTG and those with POAG had received more than one outpatient or discharge diagnosis in the ophthalmology department, and they had been observed for at least 180 days after 2008 before receiving their diagnosis of either POAG or NTG. Among them, patients with NTG and POAG diagnosis have also been excluded. Finally, the case group and control group contained 1006 and 2533 patients, respectively (Figure 1). Patients with missing data included patients who had died before the age of 20 years and those with missing values.

### 2.3. Collection of PM_2.5_ Concentration Data

The Ambient Air Quality Monitoring Network (AQMN) uses an oscillating microbalance (R&P 1400, Rupprecht and Patashnick, New York, NY, USA) to measure the concentrations of air pollutants, including PM_2.5_, PM_10_, NO, CO, etc., in hourly measurements. This information is provided by the Environmental Protection Agency, Executive Yuan. This study protocol uses months as the unit of time measurement. The standard of measurement is based on the site exposure level in the patient′s residential area; or the patient is based on the nearest station and averaged (if the patient does not live in a station). We excluded missing values based on the following criteria: during the observation period, data were excluded if daily observations contained more than 8 h of missing values, or monthly observations contained 10 days of missing values. The average daily exposure is calculated by dividing the daily exposure by 24 h, and the average monthly exposure is calculated by multiplying the average daily exposure by the number of days in the month. We observed exposure to PM2.5 for a 5-year period from 2008 to 2013 to study the effects of the development of NTG or POAG, and the loss of PM2.5 during this period was relatively small [14,19]. We used AQMN data to estimate monthly mean PM2.5 concentrations to examine differences in exposure–response relationships between NTG and POAG. Interquartile ranges are not affected by abnormally high or low values, and can be particularly useful when data are distributed asymmetrically [24,25]. It is also possible to calculate the range of alternative segments based on the data, and many research publications use the interquartile range as the standard for concentration grouping [25,26,27]. The observation period data are divided into four groups: <25th percentile (Q1), <617 μg/mm^3^; 25–50th percentile (Q2), 617–1297 μg/mm^3^; 50–75th percentile (Q3), 1297–2113 μg/mm^3^; >75th percentile (Q4), >2113 μg/mm^3^.

### 2.4. Comorbidities

Our study used the following comorbidities [19,28,29], all of which were associated with NTG or POAG, defined as at least 3 outpatient records or 1 hospitalization record prior to the date of diagnosis of NTG or POAG. In this experiment, to the confounders of physiological health adjustment, a total of 26 comorbidities were added as interference factors: arterial hypertension (ICD-9-CM: 401.x), hypotension (ICD-9-CM: 458.9), ischemic heart disease (ICD-9-CM: 434.11), sleep disturbances (ICD-9-CM: 780.57), ischemic stroke (ICD-9-CM: 434.11), Alzheimer’s disease (ICD-9-CM: 331.0), diabetes (ICD-9-CM: 250.x), Parkinson’s disease (ICD-9-CM: 332.x), congestive heart failure (ICD-9-CM: 428.x), peripheral vascular disease (ICD-9-CM: 433.x), atrial fibrillation (ICD-9-CM: 427.31), headaches (ICD-9-CM: 784), migraines (ICD-9-CM: 346), epilepsy and recurrent (ICD-9-CM: 345), rheumatoid arthritis (ICD9: 714.0), systemic lupus erythematosus (ICD-9-CM: 710.0), chronic kidney disease (ICD9: 585), hepatitis B (ICD-9-CM: 070.2, 070.3, V02.61), tuberculosis (ICD-9-CM: 010.x–017.x.), peptic ulcer (ICD-9-CM: 533), depression (ICD-9-CM: 311), malignant disease (ICD-9-CM: 14x–23x), allergic rhinitis (ICD-9-CM: 477.9), allergic conjunctivitis (ICD-9-CM: 372.14), atopic dermatitis (ICD-9-CM: 372.14), and fluid-, electrolyte-, and acid-based disorders (ICD-9-CM: 276.x).

### 2.5. Statistical Analysis

We used known confounding factors to enhance the comparability and reliability of the two groups, and adjusted for multiple variables to make the results closer to the real situation. Here, the chi-square test was used for the comparison of categorical variables, such as sex, income, and urbanization level, between patients with NTG and POAG, and a Shapiro–Wilk test was used to analyze abnormal distribution (*p* < 0.05). A Wilcoxon rank-sum test calculated the differences in continuous variables between the groups. For the regression analysis of secondary outcomes in the context of case–control sampling, logit linking is usually used to construct binary data. A simple set of estimation equations is described for inference, and a possibly more efficient approach is given. The framework also provides a formal rationale for including status as a covariate in regression models to allow for studies when the case–control disease is rare and does not require distributional assumptions [30]. We used a multivariate logic regression model to assess the risk posed to each participant. The results are expressed in terms of odds ratio (OR) and 95% CI. We used SAS 9.4 software for the data analysis, and *p* < 0.05 was considered statistically significant.

## 3. Results

The data in Table 1 indicate that no statistical difference between the patients with NTG and POAG was evident with regard to sex, age, and urbanization level. Regarding income, the majority of patients with NTG (55.77%) and POAG (61.11%) had low incomes, and the prevalence of low incomes between the two groups was significantly different. For comorbidities, no significant differences in the prevalence of arterial hypertension, sleep disturbances, diabetes, atrial fibrillation, peptic ulcer, or allergic rhinitis were identified. The proportions of sleep disturbances, atrial fibrillation, peptic ulcers, and allergic rhinitis were higher in patients with NTG (33.6%, 2.68%, 26.84%, and 28.63%, respectively), whereas the proportions of arterial hypertension and diabetes were higher in patients with POAG (39.01% and 26.69%, respectively).

Table 2 presents the distribution of PM_2.5_ exposure levels in the NTG and POAG groups. For all the participants, the median PM_2.5_ level was 1297.1 μg/m^3^, and the average PM_2.5_ level was 1366.8 (standard deviation (SD): 862.8) μg/m^3^. At the PM_2.5_ Q1 level, 662 patients with POAG and 229 with NTG were identified (median PM_2.5_ = 306.9 μg/m^3^, average PM_2.5_ [SD] = 310.5 [179.1] μg/m^3^). At the PM_2.5_ Q2 level, 644 patients had POAG and 230 had NTG (median = 895.3 μg/m^3^, average [SD] = 924.1 [199.5] μg/m^3^). At the PM2.5 Q3 level, 621 patients had POAG and 267 had NTG (median = 1693.2 μg/m^3^, average [SD] 1703.9 [243.5] μg/m^3^). At the PM_2.5_ Q4 level, 606 patients had POAG and 280 had NTG (median = 2495.7 μg/m^3^, average [SD] = 2528.0 [253.9] μg/m^3^). The increase in odds ratio for each PM_2.5_ exposure level when using univariate logistic regression was 1.111 (95% CI 1.041–1.187). Compared with the PM_2.5_ Q1 level, the OR of the PM_2.5_ Q2 level was found to be 1.032 by univariate logistic regression (95% CI 0.835–1.277), the PM_2.5_ Q3 level was 1.243 by univariate logistic regression (95% CI 1.010–1.530, *p* < 0.05), and the PM_2.5_ Q4 level was 1.336 by univariate logistic regression (95% CI 1.087–1.642, *p* < 0.05). Figure 2 presents the risk of NTG probability in patients who are exposed to PM2.5 concentrations (μg/mm^3^). The logistic regression curve adjusted for sex, age, low income, season and comorbidities. The area under the receiver operating characteristic (ROC) curve is 0.6088. As exposure to PM_2.5_ increased, the risk of NTG increased significantly (*p* < 0.0001).

Table 3 shows the risk of confounding variables for NTG development. Multiple logistic regression was used to analyze NTG patients for POAG. Compared with the PM_2.5_ Q1 level, the OR of the PM_2.5_ Q2 level was 1.009 (95% CI 0.812–1.254), that of the PM_2.5_ Q3 level was 1.241 (95% CI 1.241–1.537, *p* < 0.05),the PM_2.5_ Q4 level was 1.246 (95% CI 1.008–1.539, *p* < 0.05), and P-trend of the PM_2.5_ exposure level was 0.0117. In terms of the low income factor, the OR of low-income individuals was 0.828 (95% CI: 0.708–0.969). In terms of the urbanization level, compared with moderately urbanized individuals, the OR of highly urbanized individuals was 1.303 (95% CI: 1.083–1.566). For comorbidities, the OR of arterial hypertension was 0.704 (95% CI: 0.583–0.849), the OR of sleep disturbances was 1.274 (95% CI: 1.070–1.516), the OR of atrial fibrillation was 1.890 (95% CI: 1.121–3.187), the OR of peptic ulcer was 1.288 (95% CI: 1.069–1.551), and the OR of allergic rhinitis was 1.427 (95% CI: 1.154–1.764). These data indicate that as PM_2.5_ levels increase, the risk of NTG increases compared to POAG.

## 4. Discussion

In this study, we investigated the relationship between basic data, different comorbidities, and PM_2.5_ exposure for patients with POAG and NTG. Our research reveals that, compared with POAG, the risk of developing NTG is more closely associated with PM_2.5_ exposure, and PM_2.5_ has a concentration–dose effect.

Elevated IOP is a key risk factor for POAG, but many patients with POAG have normal IOP measurements. This condition is called NTG [31]. In NTG, optic disc cupping and VF defects usually occur without elevated IOP. Studies have shown that the morphology of the optic nerve head in patients with NTG is slightly different from that of patients with POAG [32,33]. Other studies have shown that in an age-matched linear combination model, in which iNPH/NTG + patients were analyzed, the mean cupping depth of the optic disc was significantly higher than that in iNPH+/NTG+ patients, i.e., given the extensive remodeling of the optic nerve head and progress of the disease [33,34]. Therefore, we believe that the pathophysiological pathway of NTG may differ from the pathophysiological pathway of POAG. The risk of PM_2.5_ exposure would therefore also differ.

The development of glaucoma can be attributed to external factors, including income, environmental factors, and socioeconomic development [35,36], as well as internal factors such as sex, age, family history, race, and high IOP [35,37]. Studies have noted that women are at higher risk of POAG, and POAG causes greater financial burden [38,39]. Patients with NTG are often older and tend to be female [40]. Our results indicate that the age of onset and differences in sex between POAG and NTG are nonsignificant. In South Korea, between 2004 and 2013, low- and middle-income groups were identified as being at greater risk of POAG [41]. However, some studies have noted that no obvious statistically significant association exists between high income and education level and glaucoma [42]. This may be because of the different pathological mechanisms of POAG and NTG. Our research indicates that low-income patients are at higher risk of POAG than NTG, and relatively high-income patients are at relatively high risk of NTG. We also noted that people living in highly urbanized areas were at higher risk of NTG. Research indicates that patients with POAG in rural hospitals have more advanced disease characteristics. Nearly 60% of patients in rural areas have an estimated cup to disk ratio >0.9, compared with only 20% in urban areas [43]. A WHO study noted that people living in low- and middle-income countries in Southeast Asia and the Western Pacific have 90% greater exposure to outdoor air pollution [44,45]. Outdoor PM_2.5_ components mainly come from carbon residue, human activity, and combustion-powered motor vehicles. These studies confirm that high-polluting areas are mainly concentrated in highly urbanized environments, which has a comprehensive impact on PM_2.5_ levels [19,46].

Our research indicates that sleep disturbance, atrial fibrillation, peptic ulcers, and allergic rhinitis are significantly associated with a higher risk of NTG. Previous studies have demonstrated that Alzheimer disease, vascular disease, and obstructive sleep apnoea (OSA)–hypopnea syndrome are risk factors for NTG [47,48]. Sleep disordered breathing may also be a risk factor for NTG. Various physiological factors caused by breathing-related sleep interference may play a key role in the pathogenesis of this optic neuropathy [49]. Thus, sleeping in a supine position at night may be involved in this multifactorial disease. Considerable evidence exists that in the supine position, IOP is elevated, and this is more pronounced in patients with NTG [50,51]. In both POAG and NTG, the disease progresses even if IOP is within the normal range. The prevalence of POAG and NTG is increased in patients affected by OSA, and vice versa [52]. Other studies indicate that the percentage of patients with NTG and sleep apnoea (23.07%) is higher than that of patients with POAG and sleep apnoea (3.7%), but the difference has not been shown to be statistically significant [53], and no correlation has been revealed between the prevalence of sleep disorders and the period during which patients with VF injury or POAG exhibit maximum IOP [54].

The vascular risk factors of patients with NTG are clearer. Despite a decrease in IOP, approximately 20% of patients with NTG continue to experience progressive VF deterioration [55], and many patients with VF loss have no obvious symptoms of glaucoma [56], which may aggravate the severity of VF loss in patients with NTG. Vasospasm is a vascular risk factor for NTG, and it also impairs the automatic regulation of retinal circulation [57,58]. Experiments in mice also showed that PM_2.5_-induced TGF-β1 expression was increased at higher doses of protein and mRNA levels, a mechanism for inducing heart disease [14]. This study also revealed that compared with other glaucoma patient groups, patients with NTG have a higher prevalence of cardiovascular disease [57]. A considerable body of data has confirmed the correlation between glaucoma and *Helicobacter pylori* infection [59,60,61], but other studies consider this to be controversial [62,63]. One study suggested that *H. pylori* infection may be related to an increased risk of NTG, and that this infection may play a role in the occurrence or progression of NTG as a secondary aggravating factor [64]. The essential difference between NTG and POAG may be genetic variation [65]. One study demonstrated a positive correlation between allergic rhinitis and patients with glaucoma [66], revealing that patients with allergic rhinitis and glaucoma have higher levels of nitric oxide [67,68], which may increase endothelin-1 levels, potentially explaining the relationship between allergic rhinitis and NTG in glaucoma [69]. Patients with NTG have lower circadian blood pressure parameters, which may reduce their optic nerve perfusion and cause VF damage [70]. Our results demonstrate that arterial hypertension increases the risk of POAG.

One limitation of our study is the lack of laboratory data; collecting biochemical indicators, such as patient serum values, was not possible. However, after adjusting for the comorbidity confounding factors, we believe this lack of information does not affect our results. Second, the NHIRD does not provide information on the severity of vision or VF defects, or provide optical coherence tomography images. Therefore, we could not assess whether an increase in PM_2.5_ concentrations affects the severity of NTG. Third, the NHI database does not record detailed information on the daily habits and family histories of patients, such as eating habits, smoking, and drinking. However, because of the regression adjustment and a rigorous research design, this likely had little effect on the accuracy of our results.

In summary, our population-based study is the first to use a large claims database to compare PM_2.5_ exposure levels with the relationship between NTG and POAG cohorts. A potential explanation for the close relationship between medical disease and NTG risk should be further explored. Based on this research, we can conclude that compared with POAG, the risk of developing NTG is more closely associated with PM_2.5_ exposure. In the future, when diagnosing and treating NTG, air pollution should be considered.

## Figures and Tables

**Figure 1 ijerph-19-04224-f001:**
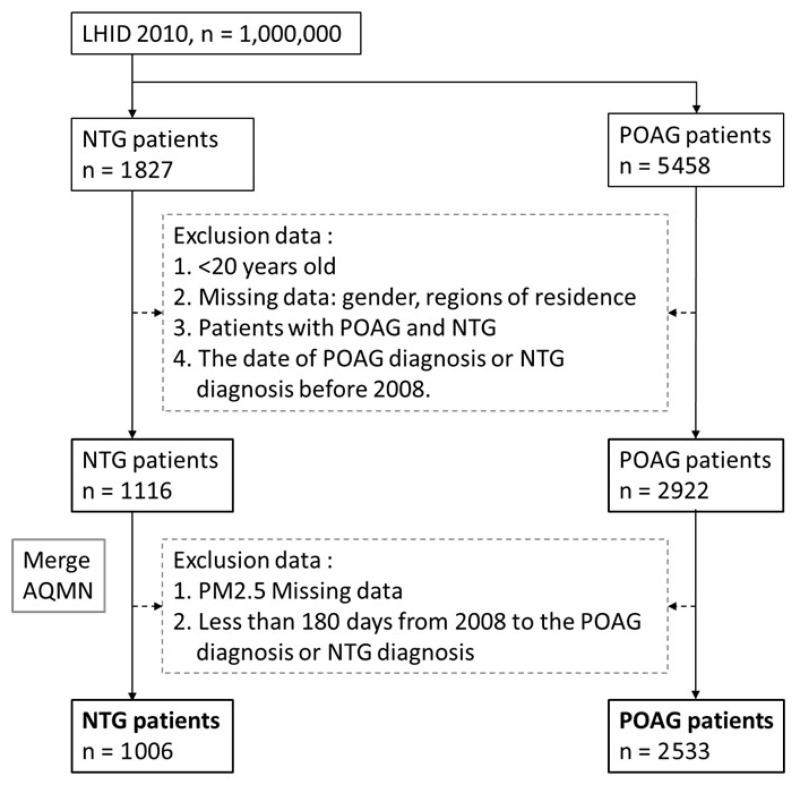
Flowchart of NTG and POAG.

**Figure 2 ijerph-19-04224-f002:**
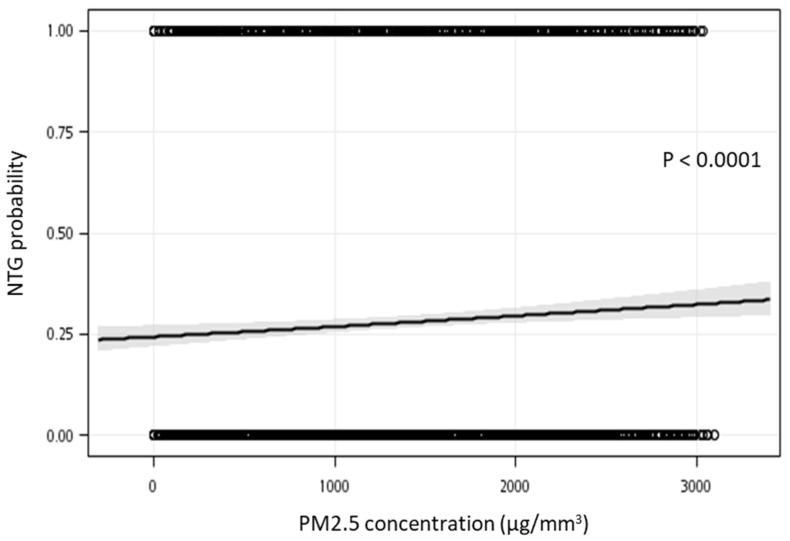
Risk of NTG probability in patients who are exposed to PM_2.5_ concentrations (μg/mm^3^), adjusted for sex, age, low income, season and comorbidities. The area under the curve of ROC is 0.6088.

**Table 1 ijerph-19-04224-t001:** Baseline characteristics of participants of POAG and NTG.

		POAG			NTG			*p*-Value
		(n = 2533)			(n = 1006)			
Gender								
	Female	1255	(49.55%)		494	(49.11%)		0.8131
	Male	1278	(50.45%)		512	(50.89%)		
Age								
	20–34 years old	339	(13.38%)		139	(13.82%)		0.2776
	35–49 years old	546	(21.56%)		216	(21.47%)		
	50–64 years old	797	(31.46%)		344	(34.19%)		
	65 years old and over	851	(33.60%)		307	(30.52%)		
	Mean ± SD	55.27 ± 18.30			55.22 ± 17.70			0.9341
Low income								
	Yes	1548	(61.11%)		561	(55.77%)		0.0035
	No	985	(38.89%)		445	(44.23%)		
Urbanization level								
	Highly urbanized	947	(37.39%)		429	(42.64%)		0.0564
	Moderate urbanization	785	(30.99%)		271	(26.94%)		
	Emerging town	365	(14.41%)		128	(12.72%)		
	General town	277	(10.94%)		117	(11.63%)		
	Aged Township	39	(1.54%)		11	(1.09%)		
	Agricultural town	69	(2.72%)		31	(3.08%)		
	Remote township	51	(2.01%)		19	(1.89%)		
Comorbidities								
	Arterial hypertension	988	(39.01%)		321	(31.91%)		<0.0001
	Hypotension	13	(0.51%)		8	(0.8%)		0.3245
	Ischemic heart disease	50	(1.97%)		15	(1.49%)		0.3345
	Sleep disturbances	723	(28.54%)		338	(33.6%)		0.0031
	Ischemic stroke	37	(1.46%)		14	(1.39%)		0.8764
	Alzheimer disease	3	(0.12%)		1	(0.1%)		0.8792
	Diabetes	676	(26.69%)		219	(21.77%)		0.0024
	Parkinson’s disease	49	(1.93%)		12	(1.19%)		0.1263
	Coronary heart disease	514	(20.29%)		195	(19.38%)		0.5425
	Peripheral artery disease	51	(2.01%)		17	(1.69%)		0.5271
	Atrial fibrillation	40	(1.58%)		27	(2.68%)		0.0296
	Headaches	904	(35.69%)		365	(36.28%)		0.7399
	Migraines	91	(3.59%)		46	(4.57%)		0.1728
	Epilepsy and recurre	31	(1.22%)		12	(1.19%)		0.9395
	Rheumatoid arthritis	54	(2.13%)		21	(2.09%)		0.9341
	Systemic lupus erythematosus	2	(0.08%)		1	(0.1%)		0.8505
	Chronic kidney disease	90	(3.55%)		33	(3.28%)		0.6894
	Hepatitis B	107	(4.22%)		44	(4.37%)		0.8426
	Fluid Electrolyte Acid	27	(1.07%)		10	(0.99%)		0.8496
	Tuberculosis	29	(1.14%)		11	(1.09%)		0.8961
	Peptic ulcer	568	(22.42%)		270	(26.84%)		0.0053
	Depression	60	(2.37%)		18	(1.79%)		0.2896
	Malignant disease	242	(9.55%)		109	(10.83%)		0.2501
	Allergic rhinitis	573	(22.62%)		288	(28.63%)		0.0002
	Allergic conjunctivitis	98	(3.87%)		38	(3.78%)		0.8983
	Atopic dermatitis	459	(18.12%)		198	(19.68%)		0.2813

Basic demographic characteristics.

**Table 2 ijerph-19-04224-t002:** Logistic regression analysis of particulate matter (PM)2.5 level between NTG and POAG.

		n (%)				Distribution of PM_2.5_ (μg/m^3^)				Odd Ratio (95% CI)
		POAG		NTG		Mean (SD)	Q1	Median	Q3	Univariate
Total participants PM_2.5_ Exposure level increase										
		2533		1006		1366.8 (862.8)	617.6	1297.1	2113.4	1.111 (1.041–1.187)
PM_2.5_ Exposure Level (reference: Q1 level)										
	PM_2.5_ Q1 level	662	(26.14%)	229	(22.76%)	310.5 (179.1)	163.7	306.9	455.6	Reference
	PM_2.5_ Q2 level	644	(25.42%)	230	(22.86%)	924.1 (199.5)	754.5	895.3	1101.6	1.032 (0.835–1.277)
	PM_2.5_ Q3 level	621	(24.52%)	267	(26.54%)	1703.9 (243.5)	1491.6	1693.2	1925.1	1.243 (1.010–1.530)
	PM_2.5_ Q4 level	606	(23.92%)	280	(27.83%)	2528.0 (253.9)	2301.5	2495.7	2730.3	1.336 (1.087–1.642)

Abbreviation: CI: confidence interval; POAG: primary open-angle glaucoma; NTG, normal tension glaucoma; SD: standard deviation.

**Table 3 ijerph-19-04224-t003:** Logic regression analysis PM_2.5_ level between NTG and POAG.

		NTG	*P*
		Adjusted OR (95%CI)	
PM_2.5_ Exposure Level (reference: Q1 level)			
	PM_2.5_ Q2 level	1.009 (0.812–1.254)	0.9357
	PM_2.5_ Q3 level	**1.241 (1.002–1.537)**	**0.0478**
	PM_2.5_ Q4 level	**1.246 (1.008–1.539)**	**0.0423**
Gender (reference: female)			
	Male	1.005 (0.862–1.172)	0.9471
Age (reference: 20–34 years old)			
	35–49 years old	0.886 (0.679–1.156)	0.3727
	50–64 years old	1.049 (0.810–1.360)	0.7156
	65 years old and over	0.919 (0.691–1.221)	0.5594
Low-income (refernce: No)			
	Yes	**0.828 (0.708–0.969)**	**0.0188**
Urbanization level (reference: Moderate urbanization)			
	Highly urbanized	**1.303 (1.083–1.566)**	**0.0049**
	Emerging town	1.055 (0.823–1.354)	0.6720
	General town	1.251 (0.961–1.630)	0.0963
	Aged Township	0.847 (0.420–1.710)	0.6440
	Agricultural town	1.420 (0.895–2.253)	0.1364
	Remote township	0.998 (0.571–1.744)	0.9946
Comorbidities (reference: without)			
	Arterial hypertension	**0.704 (0.583–0.849)**	**0.0002**
	Hypotension	1.579 (0.632–3.943)	0.3277
	Ischemic heart disease	0.740 (0.400–1.369)	0.3372
	Sleep disturbances	**1.274 (1.070–1.516)**	**0.0065**
	Ischemic stroke	1.068 (0.561–2.035)	0.8406
	Alzheimer disease	0.949 (0.092–9.749)	0.9649
	Diabetes	0.835 (0.689–1.012)	0.0656
	Parkinson’s disease	0.589 (0.304–1.141)	0.1165
	Coronary heart disease	1.044 (0.834–1.306)	0.7094
	Peripheral artery disease	0.892 (0.503–1.583)	0.6965
	Atrial fibrillation	**1.890 (1.121–3.187)**	**0.0170**
	Headaches	0.938 (0.793–1.111)	0.4590
	Migraines	1.135 (0.774–1.664)	0.5179
	Epilepsy and recurre	0.997 (0.502–1.979)	0.9926
	Rheumatoid arthritis	0.879 (0.512–1.509)	0.6412
	Systemic lupus erythematosus	1.189 (0.099–14.302)	0.8915
	Chronic kidney disease	1.075 (0.702–1.648)	0.7390
	Hepatitis B	0.979 (0.678–1.415)	0.9108
	Fluid Electrolyte Acid	0.906 (0.424–1.938)	0.7992
	Tuberculosis	0.968 (0.474–1.979)	0.9300
	Peptic ulcer	**1.288 (1.069–1.551)**	**0.0078**
	Depression	0.667 (0.384–1.158)	0.1500
	Malignant disease	1.145 (0.891–1.471)	0.2897
	Allergic rhinitis	**1.427 (1.154–1.764)**	**0.0010**
	Allergic conjunctivitis	0.808 (0.537–1.216)	0.3063
	Atopic dermatitis	0.868 (0.690–1.091)	0.2259

Adjustment for gender, age, low-income, urbanization level, Season, comorbidities. OR, odd ratio; CI, confidence interval.

## Data Availability

Research has limited the availability of these data. The study data were obtained from NHRID with permission from the National Health Insurance Administration of Taiwan, where permission was obtained from the authors.

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
