# Peer review of "Fine Particulate Matter Exposure Levels in Patients with Normal-Tension Glaucoma and Primary Open-Angle Glaucoma: A Population-Based Study from Taiwan"

_ijerph, 2022, doi:10.3390/ijerph19074224_

Round 1
Reviewer 1 Report
It is an interesting manuscript. However, the paper should be reviewed after comprehensive revision I have several comments about the manuscript.
- "It is hoped that in the future the clinical judgment of NTG and POAG...": Unfortunately, it is questionable whether a study that separates the two by collecting only ntg and poag patients will be of significance. Both diseases are on the same spectrum (OAG), and depending on the doctor diagnosed, the same eye can be coded as ntg or poag. Please read "Normal-tension glaucoma: is it different from primary open-angle glaucoma? Curr Opin Ophthalmol. 2008 Mar;19(2):85-8."
- This study should have checked whether the concentration of PM2.5 affects the occurrence of ntg or poag over the control group and should compare the effects. The current research method is meaningless in this manuscript.
- The authors have already published similar studies (Comparison of Medical Comorbidity between Patients with Normal-Tension Glaucoma and Primary Open-Angle Glaucoma: A Population-Based Study in Taiwan), but once again, it is necessary to think and check the meaning of the study. It is thought that patients with high pressure will be diagnosed as poag and patients with low eye pressure as ntg. How about revising this study with "pm2.5 risk factor analysis in the classification of high/low IOP in OAG patients?
- This study is a case–study. : not a case-control study?
- Table 2 is incomplete.
- Is there another significant relationship when linear regression analysis is performed without quartile analysis, or when more appropriate fitting is performed? It is doubtful whether the current p-value is a significant relationship compared to the entire group. Trend analysis can lead to more significant associations.
Author Response
We thank deeply for all the constructive and instructive comments from the reviewers. We had considered each comment seriously and appropriate revision was made according to the editor’s and reviewers’ suggestions. The revised phrases and wording in the text were highlighted in red.
- "It is hoped that in the future the clinical judgment of NTG and POAG...": Unfortunately, it is questionable whether a study that separates the two by collecting only ntg and poag patients will be of significance. Both diseases are on the same spectrum (OAG), and depending on the doctor diagnosed, the same eye can be coded as ntg or poag. Please read "Normal-tension glaucoma: is it different from primary open-angle glaucoma? Curr Opin Ophthalmol. 2008 Mar;19(2):85-8."
Response
Thank you for your friendly reminder. We have revised these sentences for identification of NTG and POAG in 2nd paragraph of introduction (line 79-82) and section of study population (line 101-103).
- This study should have checked whether the concentration of PM2.5 affects the occurrence of ntg or poag over the control group and should compare the effects. The current research method is meaningless in this manuscript.
Response
Thank you for your friendly reminder. We have revised these sentences in the 2nd paragraph of introduction (line 72-87).
- The authors have already published similar studies (Comparison of Medical Comorbidity between Patients with Normal-Tension Glaucoma and Primary Open-Angle Glaucoma: A Population-Based Study in Taiwan), but once again, it is necessary to think and check the meaning of the study. It is thought that patients with high pressure will be diagnosed as poag and patients with low eye pressure as ntg. How about revising this study with "pm2.5 risk factor analysis in the classification of high/low IOP in OAG patients?
Response
Thank you for your friendly reminder. We have revised the title of the manuscript.
- This study is a case–study. : not a case-control study?
Response
Thank you for your friendly reminder. We have revised the section of study population in the manuscript (line 101).
- Table 2 is incomplete.
Response
Thank you for your friendly reminder. We have revised the table 2 (line 548).
- Is there another significant relationship when linear regression analysis is performed without quartile analysis, or when more appropriate fitting is performed? It is doubtful whether the current p-value is a significant relationship compared to the entire group. Trend analysis can lead to more significant associations.
Response
Thank you for your friendly reminder. We have revised the paragraph of study population and statistical analysis in the section of method (line 127-131 and line 153-158). In addition, we also have added the figure 2 in the paragraph of results (line 193-196 and line 546). In previous studies have shown the Logic regression more suitable for case-control study due to the post-analysis interquartile range has less susceptible to bias, and the analysis items can be evenly distributed for the most statistically accurate analysis. The risk of PM2.5 for NTG under multivariate linear regression provided as below. The results between the Logistic and liner regression showed no difference.
Reviewer 2 Report
Please see the attachment.

Author Response
We thank deeply for all the constructive and instructive comments from the reviewers. We had considered each comment seriously and appropriate revision was made according to the editor’s and reviewers’ suggestions. The revised phrases and wording in the text were highlighted in red.
- The authors discussed the possible mechanisms that may relate the PM2.5 exposure and glaucoma by citing other studies. If possible, please at least give some discussion about using some animal studies to test the effect of PM2.5 exposure on the intraocular exposure. Whether changes in the trabecular meshwork ( a major aqueous humor drainage pathway), optic nerve head or retinal ganglion cells (RGC) function are involved in the pathology of PM2.5 exposure induced NTG. Whether there is cell death in trabecular meshwork or RGC after PM2.5 exposure; if so, what kind of cell death pathways such as apoptosis, necrosis or pyroptosis are involved. Do inflammation pathway or TGF beta signally pathway are involved?, etc.
Response
Thank you for your friendly reminder. We have revised the 1st paragraph of the introduction (line 62-67) and the 5th paragraph of the discussion. Modifications have been made in the first paragraph of the introduction and the fifth paragraph in the Discussion (line 267-269).
Some minor questions:
- Please make PM2.5 or PM2.5 consistent;
Response
Thank you for your friendly reminder. We have revised these worlds in the manuscript.
- From line 54, you may change “normal intraocular pressure glaucoma (NTG)” to “normal-tension glaucoma (NTG), also known as normal intraocular pressure glaucoma”;
It has been modified.
Response
Thank you for your friendly reminder. We have revised the sentence (line 53-54).
- From line 65, “(such as stroke and ischemia) Related to increased risk of sexual heart disease)”, the authors need to check whether it is right?
Response
Thank you for your friendly reminder. We have revised the sentence (line 71).
- From line 241, please change “glaucoma 56 “ to “glaucoma [56].
Response
Thank you for your friendly reminder. We have revised the worlds (line 276).
Round 2
Reviewer 1 Report
The manuscript has been improved but need some minor revisions.
Watch out for uppercase and lowercase letters. Looking at the current research direction, the previous title seems to fit better. - "Fine Particulate Matter Exposure Levels in Patients With Normal-Tension Glaucoma and Primary Open-Angle Glaucoma: A Population-Based Study From Taiwan"
I dont know the meaning of the following sentence: The ROC curve is 0.6088.
"As exposure to PM2.5 increases, the risk of NTG increases." -> "As exposure to PM2.5 increases, the risk of NTG increases significantly (P < 0.0001)."?
the trend of the p -> P?
Author Response
We thank deeply for all the constructive and instructive comments from the reviewers. We had considered each comment seriously and appropriate revision was made according to the reviewers’ suggestions. The revised phrases and wording in the text were highlighted in red.
1. Watch out for uppercase and lowercase letters. Looking at the current research direction, the previous title seems to fit better. - "Fine Particulate Matter Exposure Levels in Patients With Normal-Tension Glaucoma and Primary Open-Angle Glaucoma: A Population-Based Study From Taiwan"
Response
Thank you for your warm reminder. We have revised the title of the manuscript.
2. I don’t know the meaning of the following sentence: The ROC curve is 0.6088.
Response
Thank you for your friendly reminder. We have revised the sentence in the 2nd paragraph of results (line 194-195).
3. "As exposure to PM2.5 increases, the risk of NTG increases." -> "As exposure to PM2.5 increases, the risk of NTG increases significantly (P < 0.0001)."?
Response
Thank you for your friendly reminder. We have revised the sentences in the 2nd paragraph of results (line 195-196).
4. The trend of the p -> P?
Response
Thank you for your friendly reminder. We have revised the word (line 201).
Reviewer 2 Report
The authors have addressed my concerned, it is suitable for publication in the JCM.
Author Response
Thank you for your friendly suggestion. Because the manuscript is related to air pollutants, PM2.5, it is still chosen to be published in International Journal of Environmental Research and Public Health. Thanks again for your recognition.